# Exploring the Effectiveness of Problem-Based Learning in an International Undergraduate Program in Veterinary Sciences: Students’ Satisfaction, Experience and Learning

**DOI:** 10.3390/vetsci11030104

**Published:** 2024-02-27

**Authors:** Anna Broseghini, Miina Lõoke, Marta Brscic, Juliana Raffaghelli, Barbara Cardazzo, Antonella Lotti, Laura Cavicchioli, Lieta Marinelli

**Affiliations:** 1Department of Comparative Biomedicine and Food Science, Università degli Studi di Padova, Viale dell’Università 16, 35020 Legnaro, Italy; anna.broseghini@phd.unipd.it (A.B.); miina.looke@unipd.it (M.L.); barbara.cardazzo@unipd.it (B.C.); lieta.marinelli@unipd.it (L.M.); 2Department of Animal Medicine, Production and Health, Università degli Studi di Padova, Viale dell’Università 16, 35020 Legnaro, Italy; marta.brscic@unipd.it; 3Department of Philosophy, Sociology, Pedagogy and Applied Psychology, Università degli Studi di Padova, Via Beato Pellegrino 28, 35137 Padova, Italy; juliana.raffaghelli@unipd.it; 4Department of Humanities, Letters, Cultural Heritage and Educational Studies, Università di Foggia, Via Arpi 176, 71121 Foggia, Italy; antonella.lotti@unifg.it

**Keywords:** student-centered teaching, HowULearn, international students, soft skills, tutor evaluation, pedagogical planning, problem-based learning, PBL

## Abstract

**Simple Summary:**

A two-week module of problem-based learning (PBL) with a multidisciplinary approach was introduced into an international bachelor’s degree program in veterinary sciences in Italy. This is the first evaluation of such a program in Italy. Students expressed a high level of satisfaction and a positive attitude towards PBL. The students’ perceptions of their learning experiences and general life competencies improved after the two weeks of PBL. However, the students’ higher cognitive load and a need for feedback were also observed.

**Abstract:**

The systematic evaluation of the integration of problem-based learning (PBL) into educational programs in Italy is scarce and there are no published reports of its use in an Italian Bachelor of Veterinary Science degree program. This paper aims to assess the satisfaction of second-year students on an international Bachelor of Veterinary Science degree program after implementing two weeks of PBL with a multidisciplinary approach. Moreover, the impact of this methodological approach on the students’ performance and their perceptions concerning their learning experience was investigated. The results showed that students expressed a high level of satisfaction and a positive attitude towards learning through PBL. A significant increase in the perception of students’ soft skills was also found, based on self-evaluation. Moreover, a significant improvement was seen in the students’ perception of their learning and teaching experiences and general life competencies, assessed using the validated questionnaire HowULearn. Negative effects were also identified, requiring further design modification of the tutors’ feedback and pedagogical orchestration. Based on our findings, when planning bachelor’s degree programs in veterinary science, PBL modules or activities should be considered to promote active learning, engagement among students, and the improvement of problem-solving and team-working skills.

## 1. Introduction

Animal Care is an international bachelor’s degree program at the University of Padova belonging to the Department of Comparative Biomedicine and Food Science and started during the academic year of 2017/2018. The program is unique and aims to provide students with knowledge, skills, and competencies to work in settings in which animals are managed and cared for to enhance their health and welfare. The bachelor’s degree program is composed of a common two-year path (four semesters), followed by three study options which students apply for in their third year (fifth semester): Wild and Zoo Animals, Aquatic Animals, and Animals in Scientific Research. Students complete their program with a specific apprenticeship which is designed to enhance their professionalism (all details about the education offered through the Animal Care course can be found on the web page in the reference list [1]). International students represent approximately 40% of Animal Care students, thereby creating an international and multicultural learning environment.

Along with the knowledge and competencies acquired in each discipline, Animal Care students must develop several soft skills, including efficient communication with stakeholders from a range of backgrounds and contexts (from preschool to professionals), teamwork, time management, adaptability, problem-solving, leadership and interpersonal skills. Indeed, todays’ students should be able to integrate and connect ideas and perspectives across multiple disciplines and content areas to discover and propose new solutions. This is relevant for several professions, and in line with recent scientific publications [2,3,4], Animal Care graduates must develop the ability to promote and maintain healthy interpersonal relationships, receive and provide emotional support, and develop skills of self-care. These competencies are protective measures against poor mental health issues and compassion fatigue, to which animal caretakers and veterinarians are more predisposed compared to some other professions [5,6].

Evidence-based interventions in university teaching promote applied research into teaching and learning [7]. The European Commission’s recommendations encourage university faculties to experiment with new student-centered teaching and learning strategies. They also emphasize the need to support teaching communities, interdisciplinarity, and co-teaching options to promote reflective teaching practice [8]. Despite these recommendations for the modernization of higher education through active and personalized pedagogies [9], a tradition focused on research and transmission through didactic approaches prevails in Italian universities [10,11,12,13,14]. Indeed, while in the UK, US and Spain, faculty development has received considerable attention since the 1990s [15], in Italy, the theme has only recently started to emerge. Italian universities are now organizing internal units, like teaching and learning centers, to provide support to faculties [16]. At the University of Padua, it has been emphasized that such transformation requires a progressive cultural change in a respectful view of the professoriate traditions and practices in teaching and learning [10]. Therefore, institutional projects and community building are considered important strategies, along with continuing professional training. These initiatives also require careful monitoring and evaluation relating to their impact not only on faculty satisfaction and teaching practices, but also on the students’ opinions about the teaching and their preparedness [17,18]. The University of Padua has funded the implementation of innovative teaching projects [19] that aim to improve the teaching in undergraduate, master’s and single-cycle master’s degree courses. In this framework of educational improvement and innovation, the Department of Comparative Biomedicine and Food Science was funded to explore the introduction of problem-based learning (PBL) as a teaching strategy in the Animal Care bachelor’s degree.

The role of PBL in promoting the skills mentioned above has been proven in several published studies. Moallem evaluated the effects of PBL on learning outcomes, knowledge acquisition, and higher-order thinking skills. She found that PBL improves long-term knowledge retention, performance, and skill-based assessment measured by observation with clinical ratings [20]. Furthermore, PBL fosters the development of critical thinking skills, such as problem solving, analytical thinking, decision making, reasoning, argument, interpretation, synthesis, evaluation, collaboration, effective communication, and self-directed learning [21,22]. Micro-analytical measures, such as the questionnaires exploring the situational interest over time during PBL sessions, hold promise to explain how PBL supports student motivation and why group interactions have a positive effect on student motivation, interest, and learning [23]. PBL has also proved to be an excellent environment for building twenty-first century teamwork capability [24]. Consistent with this, a literature review of students’ perceptions about the development of generic skills or competencies in PBL educational environments showed that students had the clear perception that PBL improves their problem-solving and collaborative skills [25]. Problem-based learning, which is a well-defined method in several areas of higher education [26], has been previously applied in nursing and veterinary courses [27,28,29,30] with variable success; however, published reports of PBL effectiveness in Italy are rare. As a result, the systematic evaluation of PBL effectiveness in the Animal Care program was conducted at University of Padua.

The present paper refers to the assessment of impact and satisfaction in second-year Animal Care students, who received two weeks of PBL with a multidisciplinary approach. To the best of our knowledge, this is the first published study of its kind in Italy. Although PBL experiences are extensively reported worldwide, the specificity of each country’s university system shapes the way in which PBL can be implemented. This aspect significantly affects the student’s perception and experience during the learning process, even when more traditional teaching methods are also used [31]. Moreover, Animal Care offers a unique opportunity to assess this teaching method in a multicultural context. Thus, the aim of the present paper is to explore the possible application of the PBL methodology in an Italian bachelor’s degree in Animal Care and Veterinary Sciences. In particular, we wanted to evaluate how the methodology was reviewed by the students and if and how this methodology impacted their performance in terms of exam grades, teamwork skills, and general perception of their learning experience.

## 2. Materials and Methods

### 2.1. Participants

Nine faculty members and forty-two students were involved in the project. The faculty members included eight professors and a PhD student. All the professors involved held their courses during the first semester of the second year of the Bachelor of Animal Care degree program. All faculty members had previously undergone training on the PBL teaching approach, which included activities like retreats and workshops (for more details on faculty member training, see [26]). After the training, the university staff redesigned two weeks of lessons incorporating the PBL method and were trained to conduct the PBL sessions as a facilitator (hereafter referred to as a tutor).

All second-year students took part in the PBL sessions. Their average age was 22 years, 38 were female, 4 male, and 23 were Italians, and 19 were international students (8 were from European countries and 11 from non-European countries). Due to the different geographical origins of the students, the teaching and learning approaches that had characterized the students’ formative years had not been consistent, except for their first year together in the Bachelor of Animal Care degree program.

As required by the PBL approach, students were divided into four groups of ten or eleven participants each. During the two weeks of PBL lessons, each group worked independently and was supervised by the same tutor from among the nine faculty members.

### 2.2. Organization of the PBL Activities

The two weeks of PBL activities took place in November 2022 during the sixth and the seventh week of the first semester. The workflow of the PBL course is depicted in Figure 1.

The pedagogical approach for these two weeks was multidisciplinary and focused on the theme of zoonotic risk, which gave the two-week module its name. The module addressed several learning objectives and goals, details of which are available in the Appendix A. Before starting the module, each group of students attended an introductory presentation on course objectives and expectations and on the organization of PBL tutorial sessions. The first skill that the students needed to acquire was to be able to effectively use bibliographic resources. To this aim, in the week before starting the PBL module, a three-hour workshop was provided by the university library service. During the workshop, the library staff demonstrated how to carry out an effective search using the available databases (GalileoDiscovery, Google Scholar, Scopus, Web of Science, PubMed), and explained the library services for managing the bibliographic resources (i.e., document delivery, interlibrary loan, bibliographic consultation, remote connection). Following this lesson, the students’ skills were tested during a two-hour session supervised by a tutor where students were asked to analyze and solve a problem by searching scientific resources and analyzing their validity. This preliminary short PBL lesson served to introduce students to the teaching format of the upcoming PBL sessions. After this introductory lesson, students chose either to participate in the PBL project or to remain in courses taught using the traditional lecture methodology.

During the two weeks of the PBL course, students were presented with one multidisciplinary animal disease or management problem each week according to the seven-jump PBL model [32]. Each problem was started on Monday and completed on Friday. During the first session, the students reviewed the problem to clarify terms and fully understand the assignment. Then, students identified the problem and formulated hypotheses for the cause of the problem. Students then identified the topics they needed to study for the full understanding of the problem. Before the next session, each group of students was required to attend activities strictly related to the problem, which consisted of about 8 h overall for each of the two weeks. These weekly activities included one or two seminars on key concepts of the problem and one or two practical sessions, where students could gain experience with practical aspects related to the problem. In one of the two weeks, students were also asked to complete one assignment, which consisted of planning and presenting an intervention (i.e., enrichment protocol) related to the problem. The rest of the week was reserved for individual study. On Friday, during the final session, the students reported on what they had learned about the study topics and discussed the reliability and accuracy of the information resources they had consulted. The discussion then focused on comparing the hypotheses made on Monday and identifying the best strategies to solve the problem. After the final session, an online forum was available for the next week, through which students could contact experts for further clarification about the topic. Each of the PBL sessions lasted two hours, and at the end of the final session, twenty minutes were dedicated to students’ self and peer evaluation (see below). This evaluation was discussed among the students and supervised by the tutor, giving special attention to discrepancies between the self-assessments and those of the other group members, and between assessments of different group members.

### 2.3. Assessment and Evaluation of Students

Several assessment tools were used during and after the two weeks of PBL sessions (Figure 1). The questionnaire, Students’ Self and Peer Evaluation, [33] aimed to obtain information about the students’ evaluation of their own and others’ weekly contribution to group work (see Appendix A for the complete tool, Appendix A). At the end of the module, tutors evaluated students’ problem-solving ability, as well as their analytical and collaboration skills with the tutor assessment tool [34], based on grades from A to D (full questionnaire available in the Appendix A). Both tools assessed group work and student participation while the tutor assessment also contributed to the student’s final grade for the PBL module.

The student’s final grade of the module was composed of the weighted mean of the tutor evaluation (15%), assignment (15%), and the results of a written exam (70%). The written exam was taken one week after the PBL module and consisted of multiple choice, true-false, matching, and open short-answer questions on the core learning objectives in each module. Students could decide whether to accept the final grade of the module, in which case it contributed 15% to the final grade of each course of the semester.

### 2.4. Evaluation of the PBL Intervention

To assess students’ level of satisfaction with the different aspects of the PBL methodology (i.e., problems, module, tutor), students were asked to complete two questionnaires (Module and Problem Evaluation [34], and Tutor Evaluation [35]) within one day after the end of the module. In evaluating the tutor, students indicated their level of agreement (I completely agree; I agree, with some reservations; I somewhat disagree; I completely disagree) with statements regarding the perceived expertise, competence, and performance of the tutor. Through the Module and Problem Evaluation, students expressed their level of satisfaction with the modules’ and problems’ appropriateness and the group dynamics (the full questionnaires are available in the Appendix A).

The questionnaire HowULearn [36] (Appendix A) was administered to students the week before starting the PBL module, referring to the courses of the previous semester, and after the two-week PBL module. The HowULearn questionnaire was composed of several sections, assessing different aspects, namely, changes in students’ studying and learning styles and approaches (sections Studying and Learning I and II); university burnout focusing on exhaustion at university; cynicism toward the meaning of university, and sense of inadequacy at university (section Studying and Learning III); student’s perception of teaching activity organization and of requirements (section: Development of Teaching, composed of subsections organization and structure, teaching and learning, and requirements and assessment) and development of general life competencies (section: General Life Competencies). In the sections Studying and Learning I and II, Development of Teaching and General Life Competencies, students indicated their degree of agreement (I completely agree; I neither agree nor disagree; I completely disagree) for each statement of the questionnaire, whereas in the section Studying and Learning III, students were asked to choose the option that best described their situation.

### 2.5. Data Collection and Analysis

The Module and Problem Evaluation, the Tutor Evaluation, and the HowULearn questionnaires were made available online via Google Forms and the responses were only visible and downloadable by the university staff involved in the PBL module. All respondents completed the forms anonymously or, in the case of the HowULearn questionnaire, using a self-chosen nickname, to match the individual questionnaire completed before and after the PBL intervention. In addition, for the HowULearn questionnaire, students were asked to indicate if they were Italian or International to account for their previous learning background. For each statement of the Module and Problem Evaluation and Tutor Evaluation questionnaires, the percentage of students for each level of agreement was calculated. In the Tutor Evaluation, the identity of the tutor was not considered, and the evaluations given to the individual tutor were pooled.

The HowULearn questionnaire evaluates different aspects, so analysis was performed separately for the different sections. The different sections were identified by the original questionnaire [36] and Cronbach’s alpha was calculated to assess the internal consistency of the sections in our sample. Students’ agreement was calculated by averaging the scores of students across all statements of each HowULearn section by assigning a decreasing value to the different levels of agreement (I completely agree = 3; I neither agree nor disagree = 2; I completely disagree = 1). To assess if the mean students’ agreement was affected by participation in the PBL module and previous teaching and learning background, a GEE model was run for each section. All models included the self-chosen nickname as a random identifier to account for repeated measures, the average score of the section as a dependent variable, date of completion (before/after PBL), nationality (Italian/international), and their interaction as independent variables. The normal distributions of the residuals were graphically checked.

Finally, each student’s performance was assessed by various comparisons of grades. The rationale behind comparing performance was to assess if the PBL grades were in line with the previous ones. More specifically, we wanted to ensure that students’ exposure time to the completely new methodology was not too short to compromise their usual learning process. To this aim, the final grade of the PBL module was correlated and compared with the average grade of the previous year using the Spearman’s rank correlation and the Wilcoxon signed-rank test. Additionally, the average grade of the previous year was correlated and compared with the one obtained at the end of the second year using the Spearman’s rank correlation and the Wilcoxon signed-rank test. Finally, the average grades obtained at the end of the second year were compared between three cohorts of students using the Kruskal–Wallis test: the performance of the cohort of students who underwent PBL intervention was compared to two earlier cohorts of students who did not.

## 3. Results

### 3.1. Assessment and Evaluation of Students

The Self and Peer Evaluation questionnaires were completed by all students at the end of each week, resulting in an average score obtained by students of 3.5 ± 0.4 out the maximum of 5. The tutor assessment of students’ problem-solving ability, analytical and collaboration skills were on average 27.2 ± 2.1 out of a maximum of 30.

None of the students refused their final PBL grade. The average final grade of the PBL module was 24.3 ± 3.1 out of a maximum grade of 30. In comparison, the weighted average of the grades obtained by the students in their previous first year was 25.0 ± 2.9 out of a maximum grade of 30. Wilcoxon test of the two means found no difference between grades (Z = −1.40, *p* = 0.2) while the correlation between them was significant (r_s_ = 0.42, *p* = 0.008). However, the average grade obtained by the students at the end of the second year (25.5 ± 2.6) was significantly higher than the one obtained at the end of the first year (Z = −3.3, *p* < 0.001, Wilcoxon rank test). The two grades were highly correlated (r_s_ = 0.94, *p* < 0.001). There was no significant difference in the mean grades at the end of the second year between the cohort that had participated in the PBL activity compared to the two previous cohorts who did not (H = 0.73, *p* = 0.69, Kruskal–Wallis test).

### 3.2. Evaluation of the PBL Intervention

The questionnaire, Module and Problem Evaluation, was filled in by 30 students (71.4% of sample). The students’ evaluations of the module, the problems, and the group process and dynamics are presented in Figure 2. The level of satisfaction of the students was very high, as shown by a mean of 91.1 ± 8.2% of students who “completely agreed” or “agreed with some reservation” with the statements. The students disagreed most with the statement “The module was very well planned”, to which 33% of the students replied either “I somewhat disagree” or “I completely disagree”.

The questionnaire, Tutor Evaluation, was completed by 57 students (67.8%) and the results are reported in Figure 3. Students positively ranked the tutors, as 89.7 ± 8.9% of them “completely agreed” or “agreed with some reservation” with the statements. Students agreed the most (96.5% of the respondents) with “The tutor had well prepared the problems to be discussed” and “The tutor seemed to know the topics”, whereas the least agreed statement (71.9% of the respondents) was “The tutor helped me to make individual progress”.

The HowULearn questionnaire was filled in by 42 (100%) students before the PBL module, and by 40 (95,2%) students after the intervention. The sections Development of Teaching and General Life Competencies showed a reliable Cronbach’s alpha value (α = 0.74 and α = 0.76, respectively) and was analyzed separately. In contrast, the sections Studying and Learning I and II were grouped in the analysis, as the Cronbach’s alpha was higher (α = 0.66) than when considering each section separately (Studying and Learning I: α = 0.48; Studying and Learning II: α = 0.64). Figure 4 shows the results of the section Studying and Learning before and after the PBL intervention.

The average score for the sections after the PBL activity (estimated mean ± SD = 2.5 ± 0.04) was higher than the scores from before the PBL lessons (estimated mean ± SD = 2.4 ± 0.03; Wald chi-square = 8.4, *p* = 0.004, GEE). No significant difference was found between the Italian and international students (Wald chi-square = 1.5, *p* = 0.2, GEE) and the interaction between the previous two variables was also not significant (Wald chi-square = 1.8, *p* = 0.2, GEE).

Figure 5 depicts the results from the section Development of Teaching, composed of statements regarding organization and structure, teaching and learning, and requirements and assessment.

Similar to the previous sections, the average score after the PBL activity (estimated mean ± SD = 2.6 ± 0.04) was higher than the scores from before (estimated mean ± SD = 2.5 ± 0.05; Wald chi-square = 4.2, *p* = 0.04, GEE). No significant difference was observed between the Italian and international students (Wald chi-square = 2.3, *p* = 0.1, GEE) and the interaction between the previous two variables was not significant (Wald chi-square = 1.0, *p* = 0.3, GEE).

Figure 6 presents the results of the students’ answers about burnout related to studying and learning (Studying and Learning, section III), before and after the PBL activity. In both cases, the statement that describes the students’ situation the best was “I feel overwhelmed by the work related to my studies” (26.2% and 32.5%, before and after PBL, respectively). The statement which obtained the largest increase after the PBL activity was “I brood over matters related to my studies during my free time” (4.8% before, 17.5% after PBL), whereas the largest decrease was “I often have feelings of inadequacy in my studies” (16.7% before, 7.5% after PBL).

In Figure 7, the results about students’ agreement with statements about general life competencies (General Life Competencies section) before and after the two-week PBL module are reported.

The GEE model revealed a significant effect for the time of questionnaire completion (Wald chi-square = 50.3, *p* < 0.001), with the evaluations completed before the PBL sessions (estimated mean ± SD = 2.4 ± 0.07) having lower scores than the evaluations completed after the PBL module (estimated mean ± SD = 2.8 ± 0.04). Interestingly, the average scores of the international students (estimated mean ± SD = 2.7 ± 0.06) were higher than the scores of the Italian students (estimated mean ± SD = 2.5 ± 0.07) (Wald chi-square = 5.2, *p* = 0.02, GEE). The interaction between the evaluation date and nationality was not significant (Wald chi-square = 0.005, *p* = 1.0, GEE).

## 4. Discussion

Our results of a two-week PBL module implementation in the second year of an international bachelor degree program in Animal Care highlighted that the method effectively and positively influenced the students’ perceptions of their educational experience. However, negative perceptions related to the module organization and the tutor’s role were observed.

The response rate of our students was comparable with previous survey studies [37]. It is noteworthy that, overall, students expressed a high level of satisfaction with the PBL sessions. This demonstrates that the students had a positive experience combining relevant cognitive work with a positive learning experience. In addition, it was observed that the atmosphere of social learning in the groups was positive, in connection with the tutorial support during the application of the PBL steps. The satisfaction expressed for peer interaction, observed in our study, was also in agreement with previous findings in teaching areas related to natural and physical sciences [38,39,40] and nursing education [41]. The tutors, who play a key role supporting the learning process, were evaluated by the students on their subject knowledge and expertise, as were their organizational and teaching skills. About a fourth of the students sought additional individual help from the tutors. This aspect is well-studied under the lens of self-regulation theory, of which the main assumption is that self-organization, self-monitoring and self-evaluation are powerful predictors of effective learning [42]. Students with a low level of self-regulation tend to be less efficient and require stronger tutorship intervention [43]. Individual differences between the more self-regulated students and the less regulated ones should be expected and might explain the variability in the students’ perceptions of the tutors’ support on their individual progress [25]. As reported by Nguyen and collaborators [44] in a systematic review on the role of the tutor, key strategies such as clarification for using active learning and working continuously to support the students’ engagement in groups is extremely important. Published studies on active methods, and on problem-based learning specifically, have also reported the need to focus on feedback and support [40,41]. In natural sciences, the learning topics are complex and require careful intervention to avoid misconceptions and to support the groups’ organization, note taking, interaction through the students’ response system, among others [45]. In this regard, the organizational aspects of pedagogical practices [46] are essential, though not easy to implement, being linked to several contextual factors such as time and space constraints and the level of difficulty of the subject taught [47].

Despite high student satisfaction, some students experienced stress related to the PBL intervention. Active learning in general, and specifically PBL, is recognized as an intellectually complex methodology requiring planning, monitoring and negotiating the task with others. This can augment student resistance [48] and stress [49]. Therefore, methodologies such as PBL require specific intervention both at a cognitive and an emotional level, especially when the students are required to work in groups. As previously reported by Nguyen and collaborators [44], “The affective and behavioral domains differ from much of the prior research on active learning that centers on measuring cognitive gains in student learning (…) and affective, cognitive, and behavioral domains, (as) types of engagement are necessary for science learning through active methods”. Therefore, cognitive load and affective engagement must be carefully considered in the pedagogical orchestration of PBL [49]. In one study exploring stress levels in medical students in a PBL curriculum, stress levels associated with PBL were linked to their seniority and concerns about success in their future careers [50]. Despite these considerations, statements with high percentages of agreement such as “the problems sufficiently stimulated group discussion” and “the atmosphere was pleasant within the group” highlight the importance of positive social interactions with complex academic tasks [51].

This study also utilized the Helsinki University Learning (HowULearn) questionnaire to analyze the impact of PBL on learning and academic quality. While the tool was validated in the Finnish context, the authors considered its potential application in their English-taught Animal Care program with international students [52,53]. Indeed, examples of the application of the HowULearn scale in medicine and veterinary courses at Helsinki University are reported in the literature, along with a comparison of students enrolled in veterinary medicine in Finland and Italy [31]. In these reports, the students indicated levels of stress relating to overlapping course schedules and workload [31], as well as a lack of time to perform complex academic tasks [54]. We observed several positive effects obtained in the pre–post measurement sections of the HowULearn in our current study. In the Studying and Learning section, the students reported an improved perception of successfully managing difficult content and of acquiring relevant skills through the course. Moreover, the application of the PBL module decreased the general feeling of inadequacy towards the study and increased the involvement of the students in thinking about the disciplines during their free time. However, students also reported an increased perception of being overwhelmed, though it remained below 35 percent of respondents. The General Life Competencies subscale was also increased, highlighting an overall positive perception of the final impact of the PBL activity. This is consistent with the literature supporting the benefits of active methods of social learning through collaborative group activities [51] and the positive impact on self-care and wellbeing [5,6]. In addition, our findings highlight a higher perception of self-care (General Life Competencies) for international students both before and after the PBL intervention. It should be considered that international students might face more daily challenges in comparison to local students. Language, academic and relational skills training might be specifically relevant for them. It should also be considered that the international students come from self-selection (their intention to study abroad) which might determine the creation of a group of highly motivated and efficient students [55,56].

In relation to the effects of PBL intervention on students’ grades, there was no negative impact from a short period of exposure to a different didactical methodology. On the contrary, a possible positive effect on students’ performance at the end of the year emerged. Nevertheless, the performances of the students who participated in the PBL course were not different from those of previous cohorts who had not participated in the course. Although it has been hypothesized that minimal active learning interventions can have positive impacts on academic performance [57], the current results do not confirm higher performance by students’ in the PBL module compared to their previous average grades. Possibly, our PBL intervention was too short to modulate engagement, motivation and specific learning, for enhanced performance to emerge. Moreover, improvement in academic performance is expected if the measurement is closely related to the type of intervention [57], which was not the case in our study. Further, it should be noted that the comparison was based on different types of grades. The average grades of the previous year resulted from grades obtained in several completed exams, whereas the PBL module final grade included the tutor evaluation, one assignment, and a written exam evaluating a multidisciplinary approach. A better insight could be obtained by the comparisons of the average grades obtained at the end of the first year and the second year which compare data of the same type. Nevertheless, our results revealed that the average grades of students involved in PBL had indeed improved at the end of the second year. This might reflect a general change in the learning style after the PBL exposure, which is supported by our results in the Studying and Learning sections of the HowULearn questionnaire. An alternative and more likely explanation could be the general advancement in students’ career-related learning, as the average grades at the end of the second year of PBL students did not differ from the grades of the students from previous cohorts. Therefore, although we cannot claim an improvement in students’ academic performance after the PBL intervention, our results indicate that the short exposure to a new didactical methodology had no negative effect on the students’ academic performance.

The present study has limitations. Our results were acquired through comparisons before and after the PBL intervention lacking the evaluations that could be obtained through a randomized control trial (RCT) approach. This approach is only used under specific conditions in educational research [58]. Though the RCT is the strongest approach to generate evidence on the effect of a teaching strategy or method [59,60], its implementation is also context dependent. Accordingly, some researchers even highlight the unfeasibility of conducting RTCs in the field of education, where other types of research design should be considered [61]. A general claim is that RCTs often produce oversimplified general principles of causality and primarily serve descriptive purposes, offering limited contributions to theory generation or development [62,63]. Moreover, the ethical aspect calls for thinking about forms of intervention and data collection that do not affect the balance or ecology of group learning. Hence, educational research often adopts an approach based on the exploration of possible correlations and on the evaluation of student satisfaction [64]. It should also be considered that our data are limited to one cohort of students who underwent the PBL experience. Consequently, the current results need to be confirmed by comparing different cohorts in time series [65].

## 5. Conclusions

This study focused on the implementation and evaluation of the PBL methodology in the Animal Care international bachelor’s degree program. The results of the application of a two-week PBL module introduced in the second year support the feasibility of the methodology in this context and students’ satisfaction with the PBL implementation. Our findings also confirmed a positive correlation between PBL and achievement in soft skills such as group work, leadership, and problem-solving, among others, which are important for the students’ professional future.

To further ascertain an accurate analysis of the PBL effects, the impact of PBL should be explored through progressive cycles of evidence-based evaluation. Such an approach should be based on tools like the scales and survey adopted in the present study but also on diversified approaches. Data-driven analysis or interviews are examples of additional tools useful for deepening our understanding of the effects of PBL [66]. Furthermore, other studies on PBL have focused on different outcomes. For example, once study [67] reviewed the relevance of students’ satisfaction with the course in order to adopt deep learning approaches. The same aspect was investigated in another study [68] and found that despite the adoption of PBL, students were prone to surface learning. These two studies underline how diverse ways of PBL implementation might affect desirable outcomes such as deep learning. In summary, implementing pedagogies should be based on an analysis of their effectiveness and outcomes as a research activity accompanying innovations in the context for which they are planned.

## Figures and Tables

**Figure 1 vetsci-11-00104-f001:**
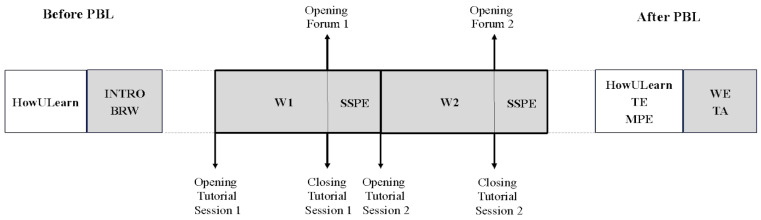
Schematic representation of the course workflow illustrating the preliminary activities aimed at introducing students to the PBL methodology (INTRO) and how to effectively search and use scientific bibliographic resources (BRW); the start and completion times of the tutorial sessions during the two weeks of PBL (W1 and W2) and the timing in which the different assessments were conducted. The white or gray background of the boxes identifies students’ evaluation of the intervention or actual didactic activities, respectively. SSPE: students’ self and peer evaluation; TE: tutor evaluation; MPE: module and problem evaluation; WE: written exam; TA: tutor assessment.

**Figure 2 vetsci-11-00104-f002:**
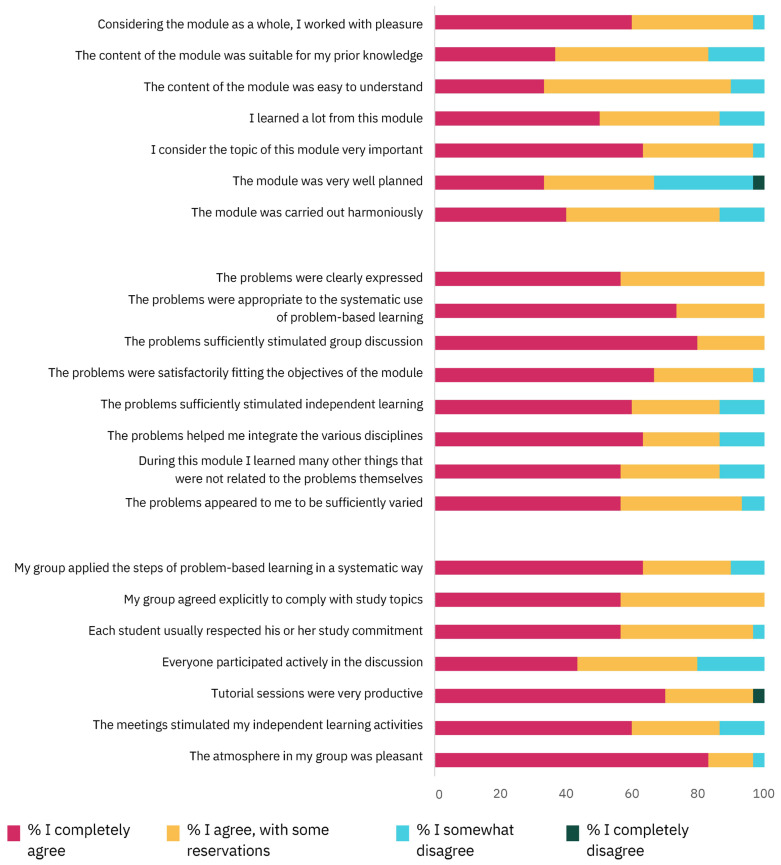
Percentage of students for each level of agreement regarding the module (first block), the problems (second block) and the group process and dynamics (third block) assessed by the Module and Problem Evaluation questionnaire (for exact percentages, see Appendix A).

**Figure 3 vetsci-11-00104-f003:**
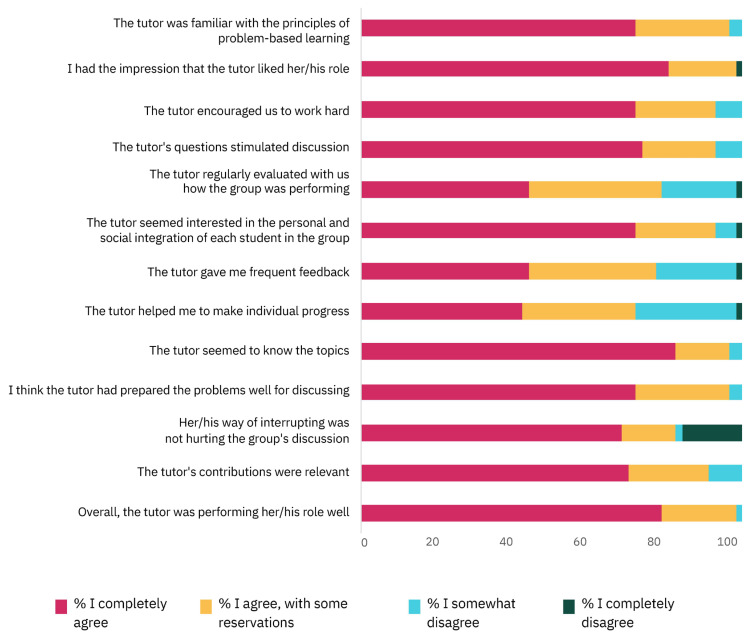
The percentage students for each level of agreement regarding the Tutor Evaluation questionnaire (for exact percentages see Appendix A).

**Figure 4 vetsci-11-00104-f004:**
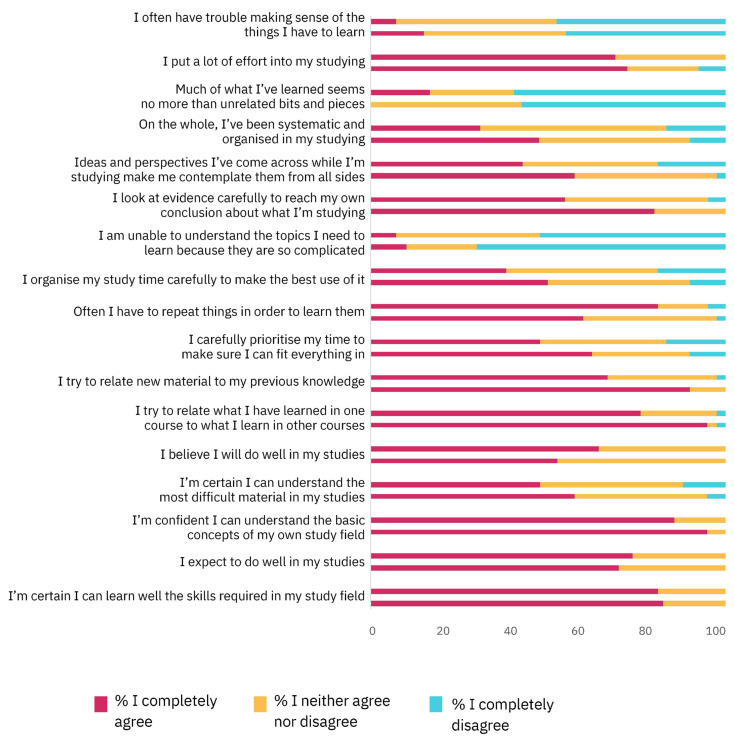
The results of the Studying and Learning I and II sections of the HowULearn questionnaire. For each statement, the upper column indicates the percentage of students for each level of agreement before attending the PBL module and the lower column indicates the percentage of students for each level of agreement after attending the module (for exact percentages see Appendix A).

**Figure 5 vetsci-11-00104-f005:**
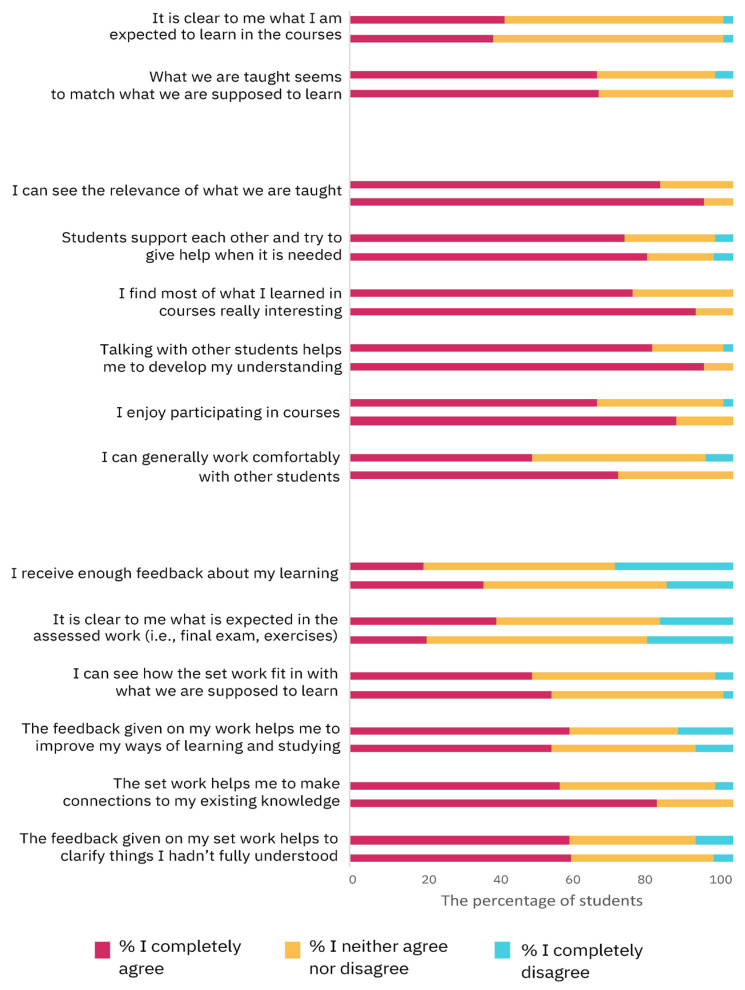
The results of the HowULearn questionnaire evaluating the students’ perception of organization and structure (first block), teaching and learning (second block), and requirements and assessments (third block) before and after undergoing the two-week PBL intervention. For each statement, the upper and the lower columns indicate the percentage of students for each level of agreement before and after attending the PBL module, respectively (for exact percentages see Appendix A).

**Figure 6 vetsci-11-00104-f006:**
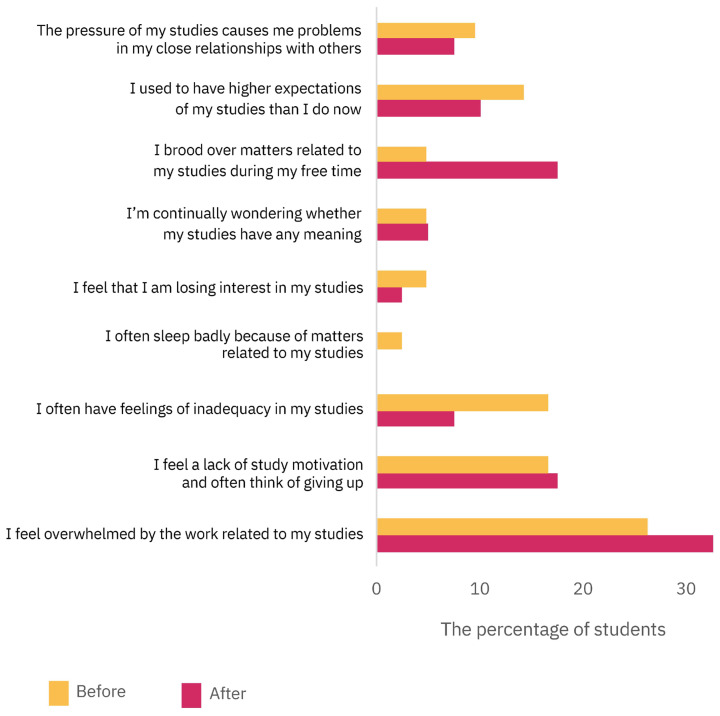
Students’ perception of burnout related to studying and learning assessed by the HowULearn questionnaire (Studying and Learning, section III). The students were asked to choose the alternative that best described their situation before and after attending the two weeks of PBL (for exact percentages, see Appendix A).

**Figure 7 vetsci-11-00104-f007:**
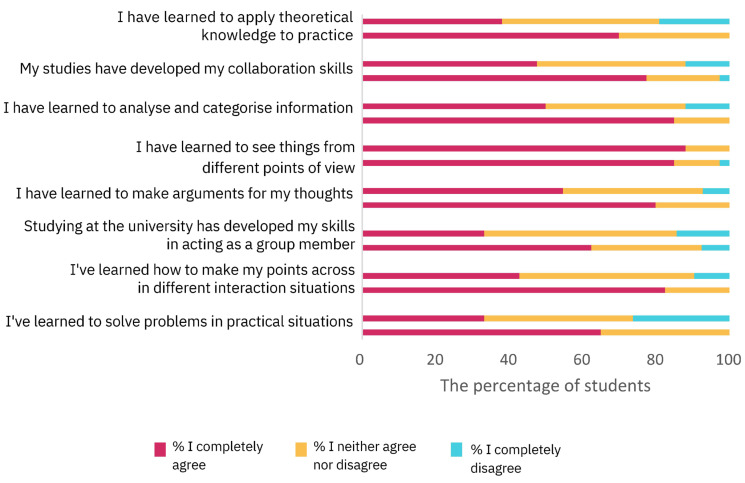
Students’ perception of the general life competencies before and after undergoing the two-week PBL module assessed by the HowULearn questionnaire (General Life Competencies section). For each statement, the upper and the lower column indicates the percentage of students for each level of agreement before and after attending the PBL module, respectively (for exact percentages, see Appendix A).

## Data Availability

Data supporting reported results can be found at: https://researchdata.cab.unipd.it/id/eprint/957 (accessed on 19 July 2023).

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
