# Peer review of "Exploring the Effectiveness of Problem-Based Learning in an International Undergraduate Program in Veterinary Sciences: Students’ Satisfaction, Experience and Learning"

_vetsci, 2024, doi:10.3390/vetsci11030104_

Round 1

Reviewer 1 Report

Comments and Suggestions for Authors

The study presents the analysis of the implementation of Problem-Based Learning (PBL) within the context of Veterinary Sciences in an international Animal Care bachelor's program. the authors observed several outcomes and challenges associated with a two-week module based on this pedagogical approach.

The authors highlighted the positive impact of PBL on students' experiences and self-confidence however, a notable portion of students expressed the need for more individualized support and feedback. 

While there were indications of improvement in certain skills specifically promoted by PBL overall academic performance did not show significant improvement. 

Line 244-245 What percentage does the module assessment contribute to the final grade for each course?

In order to assess the possible effect of the introduced methodology, the final grade in courses that include PBL must at least be compared with the students' previous average grade (not only from the previous semester). Furthermore, the average final grade results in the courses involved in the project must be compared with the results obtained in the same courses by students in previous years. 

Furthermore, the results of the correlation analysis must be thoroughly discussed because they indicate that the effect is mainly related to the personal attitude and results of each student.

Please also specify in the text which form of summative assessment is used for the courses included in the module.

Include in figure 1 the timeline to clarify how much time elapsed between the two weeks of PBL and the summative evaluation test

Lines 176-180 - Please includes Learning Outcomes on which students are assessed together with the Learning Objectives.

The Information literacy course is a general one? The same that is probably offered to all the Padua students?

Line 316-320 - please indicate your explanation for 28.5% of students that do not filled the questionnaire, this percentage can be composed by the less satisfied students?

Line 325, please explain how you obtain 57 tutor evaluation by 42 students in 4 groups. Does the tutor rotate among the groups? So, the expected questionnaires are 168, way a so limited number of answers?  

Why did you decide to use different Likert scales in the different questionnaires used?

Was the second HowULearn questionnaire completed immediately after the PBL?

Some of the results in Figure 4 related to the section Studying and Learning after PBL need to be discussed further, e.g. the increase in the percentage of students who agreed with the statement "Much of what I have learnt seems to be nothing more than unrelated bits and pieces", as well as "The ideas and perspectives I have encountered while studying make me contemplate from all sides" and many others related to the students' ability to integrate knowledge and their attitude towards deep learning.

For the processing of this section of the HowULearn results, it is difficult to understand the reason for the creation of mean values of items with opposite meanings.

Some concerns were also expressed by students about the organisation of the module, can you comment on this?

Line 461-465 The results should also be correlated with the previously published data obtained with the HowULearn questionnaire in the veterinary field (Ruohoniemi et al 2017).

Reference 18 on the T4L disciplinary projects of the University of Padua is not accessible with the given link (access on 1 December 2023); if so, please remove it and reformulate the text.

Overall, although the study acknowledges limitations in its approach and the authors make pertinent remarks on the possible application of RCTs, pre and post measures with only a two-week intervention on a small number of students (of whom only a percentage filled out the proposed questionnaires) cannot have sufficient scientific soundness without, at least, a comparison with the final results obtained by students from previous years......

Reviewer 2 Report

Comments and Suggestions for Authors

General comments

The manuscript reports in detail on student perceptions of short problem based learning activity. It shows small effect size improvements.

The authors propose that because PBL hasn’t been reported much in Italy that this is “unique” and will be of broader interest. This is not the case, as the introduction of PBL in similar didactic, traditional teacher-centred contexts has been reported extensively in other programs and other parts of the world, and in veterinary programs, including the challenges.

It will be of interest to readers as it describes a practical approach for evaluating the impacts of small learning interventions on student learning experiences. It has a useful discussion of some of the factors to consider when planning and implementing such an intervention. 

The authors should provide more information on the preparation, training and support of the faculty members who were using PBL for the first time.

The authors report on the human ethics approval was waived, as students chose to participate voluntarily. However the methods section provides information on student participants which could be identifying (and is unnecessary for the analysis) and should be generalised/removed.

The discussion over-interprets the study’s findings, which are limited to one point in time, and do not provide any indication of how student learning outcomes were impacted during the year of the study, only immediately after this module.  The opening paragraph to the discussion does not convey the relative small size, and brief time frame for the effects that are reported and the conclusion overstates the impact of the findings.

The effect sizes reported are very small. The student group size is small. The study reported one single intervention, in one year. Although some findings reached significance the discussion should reflect that these were not substantial changes in students grades or perceptions of their learning.  

The comparison of the findings of individual question from the HowULearn questionnaires taken before and after the PBL intervention show positive trends but are over interpreted. The discussion should focus on the broad groupings where significance, with a small effect size, was demonstrated.

The authors do not sufficiently acknowledge that the introduction of a different, student-centred learning activity into a didactic program has created greater stress, anxiety and loss of confidence for a small but significant group of students. These students are likely to experience dissonance, reduced motivation and to adopt a more surface, disorganised approach to learning, to their detriment. The authors could consider modifying the design and teaching approaches  used in the PBL sessions to improve the perceptions of this group of students and their approaches to learning.

The manuscript would be improved by being reduced in length to a more concise, focused discussion of the relevance to the literature and the limitations of the study.

There are numerous errors in the manuscript which need correction, usually through rewriting, to ensure clarity, precision, and ease of reading. 

There are many incorrect uses of hyphens in the middle of words. E.g. Line 421 “care-ful”

There are so many errors that a substantial rewrite is necessary to improve the clarity and precision of expression of the information and discussion.

Specific suggestions are below. Not every point requiring rewrite is indicated as there are too many.

Highlights 

Line 19.  The lack of specific evidence regarding PBL use in Veterinary science in Italy does not constitute a significant highlight or rationale for the study. PBL is used widely in the world, has been used for many years and the authors have not demonstrated any reason why this context differs to others to a degree that would require investigation.

Line 22 and 23. PBL in this study was used briefly in one component of a much larger program. The "highlights" in these lines can only be considered within this narrow context- this statement needs to be rewritten as it suggests that there was a more generalised effect.

Abstract 

Line 29 "and the impact of this methodological approach"  the subject of this clause is not clear, .... and may repeat the first part of the sentence. needs to be rewritten

Line 31-32 " significant improve-ment was observed with regard to the students’ learning and teaching experiences " - What was improved- the experience or student perceptions of it?

Line 33 "Effects such as cognitive load were identified,"  The change the authors are reporting should be clearly stated.

Line 35 "Bachelors in the area of veterinary sciences should consider" The authors mean those who are designing/implementing Bachelors programs- need to rewrite to make this clear- it is not the programs that should consider, but the designers.

Introduction 

Line 50 

"That aim" - should be plural

Line 53 - include a citation

Line 59"Animal Care students gain competences from basic disciplines to acquire specific skills and know-how to implement actions aimed at improving and promoting animal health and welfare." This sentence needs to be rewritten for clarity.

Line 62 "with figures"- this is unclear- substitute with another term eg people

LIne 71-73 For the men-tal health of animal caretakers, who are called to care for animals by vocation and are subject to compassion fatigue, these competences are reported to be preventive measures' needs to be rewritten for clarity. 

Line 92 - include citation and remove what is in brackets

Line 102 – define “Micro-analytical measures”

Line 108- remove plural “exemplars twenty-first century”

Line 112- “consolidated method”- what the authors mean by this term is not explained

Lines 233, 237, 256- identify, number the supplementary figures/tables

Lines 241, 242 – the term “incidence” is incorrect- and is unnecessary

Line 274 – “have been made available”- change tense to “were”

Line 274- it is not clear if the student nickname was also used to link HowULearn questionnaire responses to student marks/performance

 Line 388- “compilation” – the authors’ meaning in using this term is not explained

Line 399 “interfering factors”- are not explained or clear- the rest of this sentence is unclear e.g. “forms of presentation to the method”

Line 400 “It is noteworthy that…was very important” and “relevant cognitive work with the emotional aspects”- Simplify and clarify this statement

Line 403 “It has been observed”- was observed

Line 408 “about peer..” – with – It is not clear what aspect of the Deep study that is referred to

Line 409 “the result..”- which result?

Line 410 “The tutors, as a key role”-  the tutors, who have…

Line 413 “seemed to ask” -   why seemed?  

Line 417 “Knowledge and research expertise are needed to support the positive perception of stu-dents about the teacher, however.”  The meaning of this is unclear. Is it the the teacher’s knowledge and research expertise?

Line 418 “the mentioned literature”- unclear which literature is being referred to

Line 425 “Two concomitant effects relating to the attrition between” – unclear what is meant by attrition

Line 427 has not have

Line 429 “This can encompass student resistance”-  not encompass but perhaps encourage or stimulate?

Line 429 “An inverse correlation between learning and liking of methodologies such as PBL have indeed been found” – this statement is unsubstantiated and likely only correct in specific contexts

Line 439 “that centers measuring” missing on

Lines 438, 440 443 447-more inappropriate hyphens

Line 441 “orchestration”- organisation

Line 444 “As for the second effect of our intervention, we must recall that PBL, as an active learning method, might trigger.”- not a complete sentence

Line 448 commences a discussion of self-regulated learning theory, but without referencing its components of metacognition, motivation and strategic learning, and relating them to the study findings. What do the findings of this study indicate regarding the self-regulation strengths/limitations of these students? The section from line 444- 455 needs to be rewritten with greater clarity on the study findings and how they relate to the literature.

Line 460 “Hence, additional factors might explain the range observed in the response to the method in terms of satisfaction.”- rewrite for clarity and precision.

This very long paragraph should be broken up into more manageable sections.

Line 468 “Taking into account the significant results in a pre-post measurement in the scales, s” – rewrite for clarity.

Line 471 “managing well difficult contents and in the ability of achieving the a” -improve expression for clarity

Line 475-- discusses findings that have not reached significance –unnecessary

Line 483 – “along” – using

Line 485- the discussion of international students such as “It should be considered that international students face more daily challenges in comparison to local students.”- includes sweeping statement that are not referenced. It overlooks local students who might have similar or different challenges.

Line 493 “Coming to an improvement of academic performance thanks to PBL intervention, which was not confirmed by our results,”- rephrase  

Line 502 “the impact is positive.” – which impact- the sentence is unclear- what was adopted later in the exams?

Line 508 “on”- of

The very small effect size in terms of student performance should be discussed here. Consider also the validity, reliability and repeatability of the assessments.

The limitations of the study deserve a consideration of other measures of student learning experience and outcomes, and the higher education research methods that may be useful.

Line 539 “the social approach within group”- the social approach to learning?

Line 546 “coherently with – in agreement with

Line 547 “we could grasp the association between active methods and some neg-ative impacts like cognitive overload or stress”- rephrase for clarity

Line 548 “Nonetheless, the intervention highlighted the long-lasting cognitive benefits active methods overall, and specifically PBL, might of-fer.” This is not a claim that can be made from this study as it was short term, no lasting cognitive benefits were demonstrated and the effects reported were small.

Line 550 “As a matter of fact, our findings suggested a positive correlation between PBL and soft skills achievement such as group work, leadership, and problem-solving, among others, that will be important for their professional future.”-  this claim was only demonstrated to a very limited extent, as the effect size was very small.

Line 555 “Nonetheless, at the level of academic manage-ment, evaluating the impact of PBL should be considered a continuous task, which aims at comparing the students’ reactions and performance, in comparison with traditional methods.”- the meaning is unclear

Line 558 “All in all, this would open to an evidence-based approach in university teaching”- unclear- what is “this” and how would it open such an approach?

Line 559- PBL might be new in this context but is not a new pedagogy, even in veterinary education- e.g. see reference 26 from 2002

 There is a considerable literature that is directly relevant:

e.g “Students’ approaches to learning in problem-based learning: Taking into account professional behavior in the tutorial groups, self-study time, and different assessment aspects”

https://doi.org/10.1016/j.stueduc.2012.10.004

A review that addresses the research methods for investigating student approaches to learning, based on the substantial theoretical research in this field, and that discusses some of the best-accepted and validated surveys such as the Approaches to Study Inventory and the Study Process Questionnaire which the authors might find useful in reframing their discussion is:

“Using student-centred learning environments to stimulate deep approaches to learning: Factors encouraging or discouraging their effectiveness”

https://doi.org/10.1016/j.edurev.2010.06.001

Supplementary materials

The correct term is “reflect” not “mirror”.  Mirror means to precisely imitate something (reversed left to right) while to reflect is to consider and comment on one’s own experience.

Comments on the Quality of English Language

See comments above- substantial corrections and rewriting are required to improve clarity 

Reviewer 3 Report

Comments and Suggestions for Authors

It seems to me like a work that addresses a current, interesting and well-structured topic.

My review comments:

1.               What is the main question addressed by the research?

The improvement of teaching didactics in a higher education veterinary medicine course using a Problem-Based Learning strategy

2. Do you consider the topic original or relevant in the field? Does it

address a specific gap in the field?

Yes. Presents an alternative teaching model, with good results in terms acceptance and knowledge of students

3. What does it add to the subject area compared with other published

material?

 Results of a specific tested teaching methodology Problem-Based Learning

4. What specific improvements should the authors consider regarding the

methodology? What further controls should be considered?

The main concern is the temporal limitation used for collecting data (2 weeks) results of the specific  knowledge acquired by students with the PBL model in the medium and long term would be more interesting. Perhaps these aspects can be mentioned now as a limitation and presented in future works.

5. Are the conclusions consistent with the evidence and arguments presented

and do they address the main question posed?

I think the data from the study is not sufficient to make this type of conclusion “Courses in the area of veterinary sciences should therefore consider incorporating PBL modules or activities in educational programs to promote active learning, engagement among students and improvement of problem solving and team working skills”

6. Are the references appropriate?

Yes

7. Please include any additional comments on the tables and figures.

The tables do not seem necessary for the work and the figures seem important to document the MM and results.

8. Other comments

I suggest that the acronym PBL in the abstract be defined upon first use.

I also suggest that the scarcity of data 1 PBL subject module and 2 weeks be pointed out as a limitation of the work in the discussion and in the conclusions this aspect is considered so that your expectations become more moderate and are based on studies with more information

Reviewer 4 Report

Comments and Suggestions for Authors

Thank you for the submission. 

My comments below: 

Editorial suggestions: 

Abstract: 

Please check how you use acronym. For PBL, word in full should be used first (line 25) followed by acronym in brackets. Then you can continue with brackets throughout the text. Currently, you add the word in full the third time PBL is mentioned (line 28). 

Key words: 
Please check and replace key words. These should not be already included in title. 

The long paragraph from lines 75 to 120 could be broken at least in three paragraphs.  

Same (length) for paragraph from lines 426 to 492 and 493 to 532. 

References: 

Please check all of them comply with the required formatting (for example, reference 28 does not have a stop in all the abbreviation for the name of the publication). Please check them all. 

Specific comments on content 

Line 308: “None of the students refused their grade.” 

Could you please explain, not clear to me what it means. 

Figure six discusses about burnout. Burnout is also mentioned in material and methods. However, I am not clear how this relates to burnout. Or perhaps what definition of burnout was used to link the questions and the results of this particular section to burnout. I suggest including definition of burnout, as these exists and then could be easier to understand how this relate to the condition. Also, it could better guide you on the interpretation of your findings. Particularly because burnout could be linked to many factors, not only to using PBL or other methods for teaching. Timework balance in general, economic contains, personal interactions, communication (good or bad), etc.

Something I am missing in discussion is to address limitations. While the data obtained it would be sufficient to do an analysis of the impact of your teaching/learning intervention, the data obtained are closed answers, mostly in Likert scales. There was no collection of open answers in a narrative (or so it looks to me). Not suggesting you should collect it now. Instead I think this should be mentioned in your discussion, where you could comment on the robustness of your data to get the answers you want, and what else could have been done to get a deeper insight.

Comments on the Quality of English Language

In general, I think the English is good. However, there are areas of improvement in the core text, where the wording could be improved. Please re-read your submission and check the connections of the flow have the proper  grammar, providing a good flow to the text.

There was a particular area where I thought English could be improved, and this refers to figure three. However, I wonder if it is matter of translation (if the survey was not in English), but I am not certain on some of the lines there. For example: 

  • There was a sense that the tutor liked her role: What does it mean the “sense” and what does it mean the “role”? 

  • The tutor encourages us to work “hard”: is this positive or negative? Working hard may not necessarily good, it depends on how this was defined. 

  • At intervals the tutor would evaluate with us... : please check translation (if that is the case) it is correct. 

  • The tutor seemed to know the topic: it sounds too ambiguous. 

  • The way of interrupting: again, as in working hard” is this positive or negative? It depends on how this “interrupting” was defined.

Reviewer 5 Report

Comments and Suggestions for Authors

see attached file

Comments on the Quality of English Language

The paper suffers from some English issues which I have attempted to make suggestions for in more detailed comments attached. There is a tendency for overcomplicated language which becomes difficult to decipher hence a lot of the suggestions are about simplification of the message.  

Round 2

Reviewer 1 Report

Comments and Suggestions for Authors

The revisions made better clarify the dynamics of the research and correct multiple errors, but were not able to change the main criticism of the document, namely the limited nature of the intervention and the lack of evaluation of its effects on student progression and assessment results.

Author Response

Round 2: Response to reviewer 1

Reviewer 2 Report

Comments and Suggestions for Authors

The title refers an international program in veterinary science, however the introduction and abstract make it clear that this is not a registerable veterinary degree, but a degree in animal care.

The title will prove misleading for international readers, who will expect a veterinary program to produce veterinarians.  This needs to be corrected.

Comments on the Quality of English Language

While much improved, and now relatively clear, there is room for further improvement in precision of expression. 

The multiple references to students perceptions and feelings about"school" in the results section may not translate well for international readers- suggest refer to university

Author Response

Round 2: Response to reviewer 2

Reviewer 3 Report

Comments and Suggestions for Authors

I think that all the issues raised by the reviewers were introduced into the manuscript and it improved the quality, having an adequate scientific level to be published.

Author Response

Round 2: Response to reviewer 3

Reviewer 5 Report

Comments and Suggestions for Authors

The paper is much improved and more scholarly in its interpretation and critique.

I offer just a few small suggestions:

(1)    Line 434: . Active learning in general, and specifically PBL, has been characterized as complex methodologies that require planning, monitoring and negotiating the task with others.

Suggest change to something like:  ‘. Active learning in general, and specifically PBL, is recognised as an intellectually complex methodology requiring planning, monitoring and negotiating the task with others.’

(2)    Line 444: ‘According to what was observed at in medical students, stress levels associated with PBL related to their seniority and the concern of achieving success in their careers [51].’

Suggest change to ‘In one study exploring stress levels in medical students in a PBL curriculum, stress levels associated with PBL were linked to their seniority and concerns about success in their future careers [51].

(3)    Ethical Issues

AU: We understand the reviewer's comment and concern about this point, however this outcome was based on institutional decision.

I defer to the journal on this point. As a minimum I would at least like to see some acknowledgment of this issue  - just because the institution apparently doesn’t have an ethical stance on student surveys which can touch on potentially challenging wellbeing matters, it doesn’t devolve the researchers of their responsibility to be cognisant of wellbeing issues that may arise as a result of the questions/ methods they used. Perhaps they could address this in their limitations section or as another footnote?

Comments on the Quality of English Language

Improved - some minor editing will still be needed

Author Response

Round 2: Response to reviewer 5
